# Psychophysiological Responses of Humans during Seed-Sowing Activity Using Soil Inoculated with *Streptomyces rimosus*

**DOI:** 10.3390/ijerph192316275

**Published:** 2022-12-05

**Authors:** Na-Yoon Choi, Sin-Ae Park, Ye-Rim Lee, Choong Hwan Lee

**Affiliations:** 1Department of Bio and Healing Convergence, Graduate School, Konkuk University, Seoul 05029, Republic of Korea; 2Research Institute for Bioactive-Metabolome Network, Konkuk University, Seoul 05029, Republic of Korea; 3Department of Systems Biotechnology, Konkuk University, Seoul 05029, Republic of Korea; 4Department of Bioscience and Biotechnology, Konkuk University, Seoul 05029, Republic of Korea

**Keywords:** soil microorganism, electroencephalogram, serum metabolites, GC-TOF-MS, horticultural therapy, psychophysiology, gardening, geosmin, metabolite profiling

## Abstract

Electroencephalogram (EEG) responses and serum metabolite levels were used to investigate the effects of horticultural activities (seed-sowing) on the psychophysiological aspects of adults based on the presence or absence of the soil microorganism *Streptomyces rimosus*. In this case, 31 adults were subjected to seed-sowing activities using *S. rimosus* inoculated (experimental group) and medium (control group) soils. EEG was measured to analyze the resulting psychophysiological response, and blood samples (5 mL) were collected. The relative gamma power (RG), relative high beta (RHB), and SEF 50 and SEF 90 were significantly higher in the right than in the left occipital lobe (*p* < 0.05). In both occipital lobes, ratios of SMR to theta (RST), mid beta to theta (RMT), and SMR-mid beta to theta (RSMT) were high (*p* < 0.05). GC-TOF-MS-based serum metabolite analysis detected 33 metabolites. Compared to the control group, the experimental group showed a lower content of amino acids (except aspartic acid), lipids, and C6 sugar monomers after the activity (*p* < 0.05). Aminomalonic acid was decreased, and aspartic acid was increased (*p* < 0.05). This study confirmed a positive effect on improving the concentration and attention of adults when seed-sowing activity was performed using *S. rimosus*-inoculated soil.

## 1. Introduction

Globally, rapid urbanization is associated with population growth in urban environments and decreasing natural environments, and common mental disorders are more prevalent in cities [1]. Since rapid urbanization affects various factors related to modern lifestyles, an imbalance in the microbiome of the human body may occur, affecting mental disorders and immune-mediated diseases [2,3,4,5,6]. Furthermore, in contrast with the pre-industrial rural environment, the modern urban lifestyle causes a loss of direct contact with the soil, possibly impeding the exchange of microorganisms between humans and the environment [7].

Soil is the most extensive reservoir of natural microorganisms on the planet [8], and contact with non-pathogenic soil microorganisms has been reported to positively affect human health [9,10,11]. Increased exposure to various natural environments can positively affect human health by increasing microbial diversity in the human body [12]. Some of the microorganisms expressed in soil exchange hormones and immune signals between the gut and the brain, and have various beneficial effects on humans [13]. In addition, soil microorganisms and physiologically active volatile organic compounds (VOCs) derived from these microorganisms enter the human body through skin contact or the respiratory tract [14,15]. Continuous exposure to organisms such as soil microbes can trigger tolerance responses to stress [16]. However, this observation was from an experimental study on mice, and studies with human intervention are insufficient to date.

Furthermore, soil microorganisms have been studied in various ways as a role of promoting plant growth hormones. The *Streptomyces*, one of the main microorganisms present in soil, produces various physiologically active substances, plays an important role in soil fertilization, and protects roots from plant pathogens [17]. The potential of the soil microorganisms *Streptomyces* to promote plant growth has been frequently reported [18,19]. It has been shown to promote root growth and increase chlorophyll content through volatile compounds of *Streptomyces spp*. [20], and chlorophyll and proline content were significantly increased in plants inoculated with *Streptomyces* culture medium [21]. Some have been developed and applied as antibiotics applied to humans to treat diseases [22]. As such, it was found that soil microorganisms have effects related to plant growth and are also used as major substances in antibiotics used for human diseases. As in previous studies, various studies on plants of soil microorganisms have been conducted, but studies on contact with soil microorganisms or the reaction of fragrance components generated therefrom to the human body are very lacking.

Soil microorganism *Streptomyces rimosus (S. rimosus)* is a bacterial species of the genus *Streptomyces* commonly found in soil. In addition, *S. rimosus* has also been shown to release large amounts of geosmin and 2-methylisoborneol (2-MIB) [23], which are the major volatile metabolites of common earthy odor [24]. A study measuring the effect of olfactory stimulation of these components on the psychophysiological responses of adults showed that alpha wave activity increased in the brain’s frontal lobe, which positively affects emotional stability [15].

Furthermore, alternative therapies that actively use the effects of environmental substances, such as natural comfort and immunity improvement, are known to have positive effects on the body, such as sterilization, improvement of cardiopulmonary function, and lowering of blood pressure [25,26]. Another soil microbial species, *Mycobacterium vaccae (M. vaccae)*, is a non-pathogenic bacterial species belonging to *Actinomycetales*. In a recent study, using soil inoculated with *M. vaccae* [27] and *S. rimosus* [23] for adults, soil mixing was performed for 5 min and EEG and metabolic responds were analyzed. As a result, alpha wave activity increased in the occipital lobe of the brain, showing a psychophysiological stabilizing effect [27]. In addition, serotonin was found to increase significantly in blood metabolite analysis [23]. It has been reported that soil mixing activity using soil microorganisms can lower anxiety and stress and increase vitality. Based on these previous studies, volatile metabolites produced by microorganisms in the soil can affect changes in human psychophysiology and biological metabolites when introduced through the respiratory tract.

However, all of the above studies have limitations in that just one motion (soil-mixing) was repeated only for 5 min, and the adequacy of the intervention time could not be verified. In order to identify the psychophysiological mechanism according to the presence or absence of soil microorganisms, it was suggested that additional studies according to the difference in intervention time, frequency, and activity are needed [23].

Therefore, this study tried to intervene in practical horticultural activities while supplementing the limitations of previous studies, carried out seed sowing activities in 4 steps for a total of 20 min, and investigated the psychophysiological and metabolic reactions of adults during the activities. In that sense, this study is meaningful as an initial exploratory study that has progressed from the context of previous studies.

## 2. Participants and Methods

### 2.1. Participants

This study was conducted on 31 adults in their 20s to 50s (9 males, 21 females, average age 32.66 ± 10.16 years). Participants used a convenient sampling method. Participants were visited twice, and the same horticultural activity (seed-sowing) was performed. In addition, according to a study that the hand movements of the dominant and non-dominant hands differ in brain activity, only those whose dominant hand was the right hand participated in this study [28]. Those with a history of cardiovascular diseases, such as high blood pressure, heart surgery, or were taking related drugs, and cognitive impairment of psychopathological problems such as depression or panic were also excluded from the study (Table 1). In addition, considering that brain arousal components such as caffeine may affect the experimental data, fasting and smoking cessation were requested 2 h before the experiment [29]. Before the study began, the contents of the study and the method of the horticultural activity program were explained to all participants, and those who agreed to participate signed a consent form.

A questionnaire was then prepared to collect demographic information, and height, weight, and body mass index (ioi 353; Jawon Medical, Gyeongsan, Republic of Korea) were measured. A scent survey for screening test (SSS test) ranked by priority and VAS score was performed to select participants with normal olfactory function [31]. The participants were paid a small incentive (23 USD) for participation in the study. This study was conducted with the approval of the Konkuk University Institutional Bioethics Committee (7001355-202112-HR-494).

### 2.2. Experimental Environment

This study was conducted in an experimental space (180 cm × 200 cm) in the greenhouse of the Konkuk University campus in Seoul (Figure 1). In order to minimize external visual stimulation, a white cardboard was laid out in front of the desk, and ivory-colored curtains were installed on both sides. The environmental conditions of the experimental space were an average temperature of 27.3 ± 3.0 °C (O-257; DRETEC Co., Saitama, Japan), average humidity of 43.3 ± 15.8 % (O-257; DRETEC Co., Saitama, Japan), and average illuminance of 2097.9 ± 1076.8 lx (Lux Light Meter; Przemek Pardel Co., Zurich, Switzerland).

### 2.3. Preparation of the Soil Sample

A total of two types of soil were used in this study, and the common mixed materials were peat moss (2000 mL), perlite (800 mL), and water (200 mL). In order to prepare samples in a sterile state, soil samples were autoclaved at 121 °C for 15 min. Culture medium (50 mL) without microorganisms was mixed with the sterilized material and used as control soil, and 50 mL of *S. rimosus* KACC 20082. *S. rimosus* KACC 20082 was procured from the Korean Agricultural Culture Collection (KACC), and the culture medium was cultured for three days, mixed, and used in the experimental soil.

### 2.4. Seed Sowing Activity

This study selected seed sowing activities frequently used in actual horticultural treatment sites to carry out horticultural activities using soil inoculated with soil microorganisms. In addition, coated seeds were used after sterilization to block the intervention of microorganisms expressed in plants. The activity proceeded in four steps, which were mixing the soil, transferring the soil to the sowing tray, planting seeds, and watering them with a sprayer (Figure 2). The total duration of the activity was 20 min, the type of soil was randomly provided at each visit, and the contents and sequence of the activities were same.

### 2.5. Experimental Protocol

Before the start of the experiment, after explaining the experimental procedure to the participants, height, weight, and body mass index were measured. Afterward, the participant sat in the experimental space and wore a wireless electroencephalograph (Quick-20, Cognionics, Inc., San Diego, CA, USA). The seed sowing activity comprised of four stages (soil mixing, soil filling, seed sowing, and watering). Before starting the activity, the participant was asked to face the front and rest for 5 min. Next, the soil in the basin was mixed and transferred to a sowing tray. After that, the petri dish containing the seeds was opened, and the seeds were sown individually in the tray and finally watered with a sprayer. Each step was performed for 5 min, and the activity was performed for 20 min. Blood samples (5 mL) were collected from each participant before and after the activity. The experiment was conducted through a blind experiment and a randomized crossover study method, and the total experiment time was up to 40 min (Figure 3).

### 2.6. Measurements

#### 2.6.1. Electroencephalogram (EEG) Measurement

EEG was measured to determine the psychophysiological responses of adults when performing seed-sowing activities according to the presence or absence of *S. rimosus*. The electroencephalograph uses dry electrodes to minimize the risk of electrical stimulation and has been certified for safety by the European Commission and the Federal Communications Commission [32]. Data were collected by amplifying the electrical signal measured by attaching a dry electrode to the scalp. According to the international 10–20 electrode placement system [33], the reference electrode was attached to the left earlobe (A1), and EEG electrodes were attached to the positions corresponding to the left occipital cortex (O1) and right occipital cortex (O2). Thereafter, monitoring was performed (Figure 4). A previous study on the psychophysiological response of adults to soil mixing according to the presence and absence of the soil microorganism *Mycobacterium vaccae* showed that the brain comfort index was high owing to an increase in alpha waves in the occipital lobe [32].

In this study, it was hypothesized that volatile metabolites such as geosmin and 2-MIB, which induce an earthy odor in *S. rimosus*, may have affected the olfactory stimulation during activity. In addition, it has also been reported that olfactory stimulation can improve understanding of brain activity and human central nervous system activity [34]. Therefore, in this experiment, the occipital cortex was selected and analyzed to investigate the effects of horticultural activities, using soil inoculated with or without *S. rimosus*, on the psychophysiology and metabolomic response of adults.

#### 2.6.2. Measurement of Serum Metabolites

In order to determine the metabolite levels during seed-sowing activities according to the presence or absence of *S. rimosus*, blood samples (5 mL) were collected before and after each soil activity. Blood samples were collected by trained professional nurses, placed in ice packs, and transported to the analysis site. Serum samples were separated by centrifugation at 3000 rpm for 10 min. After that, aliquots were stored in a deep freezer at −70 °C.

### 2.7. Data Analysis

#### 2.7.1. EEG Data Analysis

EEG data were analyzed using the Bio-scan (Bio-Tech, Daejeon, Republic of Korea) program, and the frequencies analyzed in this study are shown in Table 2.

The processed EEG data were subjected to a paired t-test using SPSS (version 25 for Windows; IBM, Armonk, NY, USA). The significance level was set to *p* < 0.05. In order to analyze demographic information, descriptive statistics on the mean and standard deviation of each collected item were performed using Microsoft Excel (Office 2007; Microsoft Corp., Redmond, WA, USA).

#### 2.7.2. Serum Metabolite Analysis

Serum samples collected were transferred to Konkuk University ‘M’ Research Center for analysis. Each serum sample was thawed at 4 °C. Next, 200 μL of each sample was transferred into a 2 mL Eppendorf tube, 800 μL of cold methanol and 8 μL of internal standard (2-chloro-phenylalanine, 1 mg/mL in water) were added to the tube, and the mixture was vortexed for 1 min. The mixture was homogenized for 10 min at a frequency of 30 Hz using a mixer mill (MM400; Retsch, Haan, Germany) and then sonicated for 10 min again. The treated sample was centrifuged at 13,000 rpm at 4 °C for 10 min, and the supernatant was filtered through a 0.2 µm polytetrafluoroethylene filter and completely evaporated using a speed vacuum concentrator. The final concentration of each sample was adjusted to 10 mg/mL for mass spectrometry (MS) analysis.

After collecting raw data with a GC-TOF-MS device for lysed serum samples, GC-TOF-MS data were digitized using ChromaTOF software (version 4.44, LECO Corp., St. Joseph, MI, USA) and Metalign program, and data processing such as peak selection, alignment, and baseline correction was performed [23,35]. Multivariate statistical analysis was performed on the data quantified through the Metalign program using SIMCA-P+ (Ver. 12.0) software (Umetrics, Urea, Sweden). Student’s *t*-test and PASW (Predictive Analytics SoftWare) statistical software (SPSS Inc., Chicago, IL, USA) were used for significant metabolite differences, and the significance level was set to *p* < 0.05.

## 3. Results

### 3.1. Demographic Information

The average age of the participants in this study was 32.66 ± 10.16 years, with a total of 31 patients, nine males (29.0%) and 22 females (71.0%). In addition, the average height of the participants was 165.94 ± 7.23 cm, and the average weight was 64.27 ± 11.54 kg. Therefore, the body mass index was 23.25 ± 3.23 kg·m^−2^, which was confirmed to be within the normal range according to the World Health Organization standards. Furthermore, from the self-reported olfactory function evaluation, the SSS score was 82.2 ± 10.7, and the VAS score was 7.6 ± 1.1, which belonged to the normal olfactory group (Table 3).

### 3.2. Results of EEG Responses

EEG analysis results when performing seed-sowing activities according to the presence or absence of *S. rimosus* are shown in Table 4 and are as follows.

In the RG power spectra analysis results, it was significantly higher when the activity was performed using *S. rimosus* soil (experimental group) in the right occipital lobe (*p* < 0.05). RG is a prominent indicator of cognitive processing, such as reasoning and judgment [36,37]. The RHB power spectra were significantly higher when the activity was performed using *S. rimosus* soil in the right occipital lobe (*p* < 0.05). RHB is a high indicator of excitement and tension [37]. In the RST power spectra analysis results, it was significantly higher when the activity was performed using *S. rimosus* soil in both occipital lobes (*p* < 0.05). RST is an EEG indicator that reflects attention [38]. The RMT power spectra were significantly higher in both occipital lobes when the activity was performed using *S. rimosus* soil (*p* < 0.05). RMT is involved in attentional concentration, and the higher the RMT, the higher the concentration [36,38]. In the RSMT power spectra analysis results, it was significantly higher when the activity was performed using *S. rimosus* soil in both occipital lobes (*p* < 0.05). This is known as a neurophysiological concentration index [36,37,39]. In addition, after analysis of the SEF50 and SEF90 indexes indicating the level of brain arousal, it was significantly higher when the activity was performed using *S. rimosus* soil in the right occipital lobe (*p* < 0.05).

### 3.3. Results of Serum Metabolite

The PCA analysis was performed on serum collected after each activity using soil to investigate the reaction of metabolites in blood during seed-sowing activities according to the presence or absence of *S. rimosus*. The results showed that the medium (control group) and *S. rimosus* (experimental group) soils tended to be distinctly divided. Therefore, OPLS-DA analysis was performed to evidently observe the difference between the two groups, and as a result, the medium and *S. rimosus* soil groups tended to be divided by OPLS1 [9.83%] (*p* < 0.05) (Figure 5).

The metabolite analysis was performed based on the OPLS-DA model showing significant differences. A total of 33 metabolites (one organic acid, 15 amino acids, seven carbohydrates, five lipids, three others, and two unknowns) were detected. The detected metabolites were displayed on a heat map for visualization (Figure 6), and amino acids (except aspartic acid), lipids, and C6 sugar monomers showed significantly lower content in *S. rimosus* soil than in the control soil (*p* < 0.05).

## 4. Discussion

This study was conducted to investigate the effect on the psychophysiology and metabolic response of adults when performing seed-sowing activities based on soil inoculated with the soil microorganism *Streptomyces rimosus (S. rimosus).*

*Streptomyces* emit large amounts of VOC components (geosmin and 2-MIB) that cause a general earthy odor [24]. VOCs from soil play an important role in relieving inflammation and stress, alleviating sleep disturbance, and regulating the human immune system [40,41]. Geosmin and 2-MIB belong to the terpenoid compound group. As a result of measuring the effect of olfactory stimulation of geosmin and 2-MIB on the mental and physiological health of adults, it was found to be effective for emotional stability by increasing alpha wave activity in the frontal lobe of the brain [15]. Based on these previous studies, it can be confirmed that olfactory stimulation by soil-derived VOCs also affects the central nervous system and cerebral cortex. In addition, the characteristic odor of microbial cultures of actinomycetes widely distributed in the soil is caused by volatile terpenes [42,43,44,45,46,47]. Previous studies have reported the detection of numerous volatile terpenes in *Streptomyces* [48,49,50,51,52,53,54,55,56]. The most commonly detected terpenoids in *Streptomyces*, geosmin, and 2-MIB represent volatile metabolites [48,51,57].

In this study, EEG was measured to determine psychophysiological responses when performing horticultural activities based on the presence or absence of *S. rimosus*, and physiological changes were measured through a metabolomic approach. As a result of EEG analysis, when horticultural activities were performed using soil containing *S. rimosus*, RG, related to cognitive function, RHB, related to the state of tension, and SEF50 and SEF90 indices indicating brain arousal and immersion were significantly higher in the right occipital lobe (*p* < 0.05). RST, RMT, and RSMT were high in both occipital lobes (*p* < 0.05).

In a previous study, when performing a soil mixing activity using soil microorganism, *M. vaccae* and *S.rimosus* for 5 min, the alpha wave index of the occipital lobe was significantly increased [27], and the serotonin index related to emotional stability increased [23]. Previous research results showed that brain activity and autonomic nervous system were stable, but in this study, a difference was found with a significant increase in indicators related to brain arousal and concentration. This is different in that previous studies only performed soil mixing for 5 min, but in this study, seed sowing was applied as a type of universal horticultural activity and exposed to soil microorganisms for 20 min. In addition, in this study, a significant difference was shown according to the use of *S. rimosus* inoculation soil and medium soil, which can be seen as a result of soil-derived VOC components shown in previous studies. In order to understand whether odor molecules can cause functional changes such as relaxation and excitability states in the brain, several studies are being conducted validating the effects of olfactory stimulation [58]. EEG is the most effective method to investigate electrophysiological changes in the brain in a non-invasive manner, and previous studies have used EEG to determine EEG changes during inhalation of odor molecules [59,60,61]. However, considering that the limitation of this study is the level of literature survey on soil VOCs, it is important to study the olfactory stimulation of major VOCs such as terpenes through the analysis of VOC components of *S. rimosus* in subsequent studies.

In addition, as a result of analyzing the metabolic reactions of adults during horticultural activities in the presence or absence of *S. rimosus*, a total of 33 metabolites (one organic acid, 15 amino acids, seven carbohydrates, five lipids, three others, and two unknowns) were detected. Compared to the medium soil (control group), the *S. rimosus* soil (experimental group) showed a lower content of organic acids, amino acids, and carbohydrates (*p* < 0.05), and decreased aminomalonic acid and increased aspartic acid among amino acids (*p* < 0.05). In a study that analyzed metabolites after exercise therapy in diabetic patients, the relative contents of benzoic acid, aminomalonic acid, tetrabutyl alcohol, and ribonucleic acid were significantly reduced in the diabetic exercise group compared to the control group [62]. This study indicated that the metabolomic method is a useful tool for studying the mechanisms of exercise therapy. In addition, it is thought that the physical and motor effects of horticultural activities can be evaluated because aminomalonic acid was significantly lowered from the results of this study. Li et al. [63] used gas chromatography-mass spectrometry (GC/MS) metabolite approaches combined with principal component analysis (PCA) and open partial least squares discrimination analysis (OPLS-DA) statistical analysis to detect different metabolites in peripheral blood monoclonal cells (PBMC) in depressed mice. Reduced levels of aspartic acid, glutamic acid, and glycine were observed in depressed rats compared to healthy rats (*p* < 0.05), suggesting that it could be useful for clinical diagnosis of depression of three types other than aspartic acid, a major neurotransmitter. Fu et al. [64] also showed changes in plasma concentrations of the excitatory amino acid neurotransmitters aspartic acid (Asp), glycine (Gly), and aspartic acid (Asn) in patients with depression disorder, which were significantly lower than in the control group (*p* < 0.05). In this study, when the activity was performed using *S. rimosus* soil, the level of aspartic acid among amino acids was increased, so the significance of the metabolites related to depression can be examined as a change.

## 5. Conclusions

In conclusion, brain arousal and attention index significantly improved when horticultural activity (seed-sowing) was performed using soil inoculated with the soil microorganism *S. rimosus*. In addition, it was confirmed that aminomalonic acid among metabolites decreased and aspartic acid, a neurotransmitter, tended to increase. However, this is also a part of basic research, and it is difficult to generalize in that it consists of a small population. Therefore, in future studies, it is necessary to collect data by expanding the population based on more specific and objective standards. In addition, the dominant hand in this study consisted of only right-handed participants, but in future studies, it is necessary to increase the population and compare left and right dominant participants. Additionally, based on this study, future studies need to verify the exact healing mechanism through microbial profiling and analysis of volatile organic compounds generated when soil microorganisms are mixed.

However, this study is meaningful in that it supplemented the limitations of existing studies, extended the intervention time, and attempted additional research by performing practical horticultural activities. In addition, since a significant index was newly presented in the metabolic reaction analysis, a differentiated conclusion can be drawn as a follow-up study of the existing preceding studies.

In conclusion, based on the results shown in this study, it can be used as basic data for the development and design of horticultural programs using soil. In addition, in future studies, it is necessary to investigate the scientific healing mechanism through the evaluation of effects targeting patients with specific diseases such as depression, anxiety, and cognitive disorders, and the analysis of VOC components produced by soil microorganisms.

## Figures and Tables

**Figure 1 ijerph-19-16275-f001:**
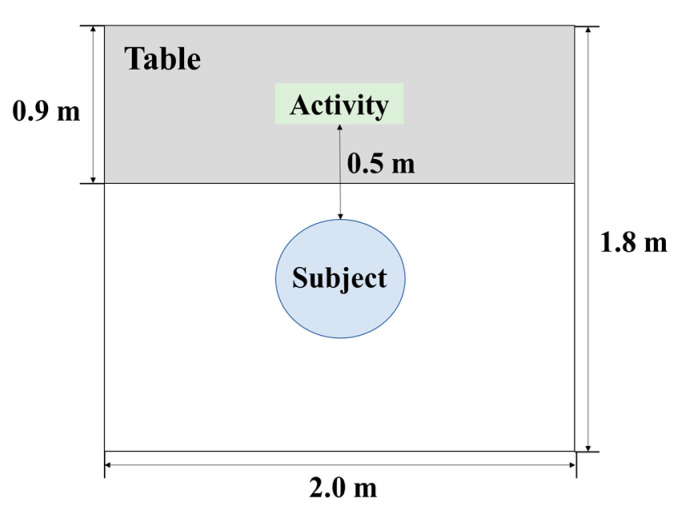
The room arrangement of the experiment.

**Figure 2 ijerph-19-16275-f002:**
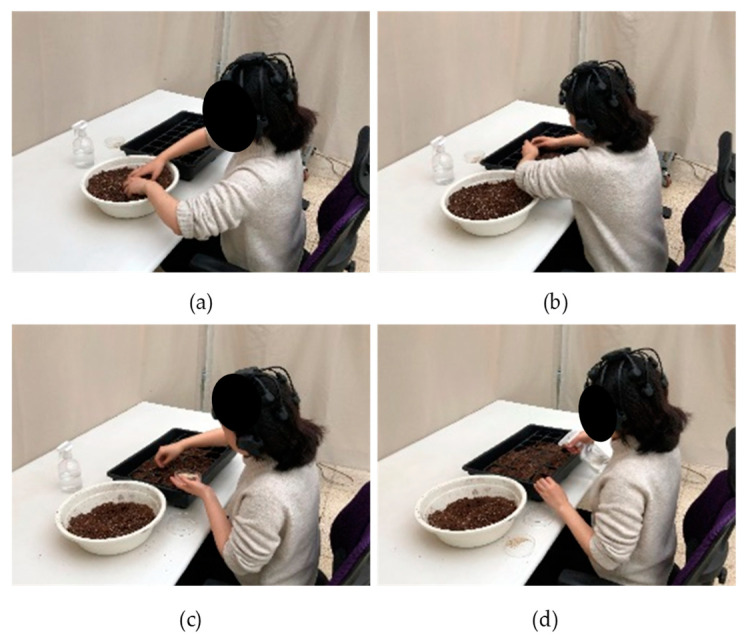
The experimental performance appearance: (**a**) mixing the soil, (**b**) putting soil in the sowing tray, (**c**) sowing seeds, and (**d**) watering with a sprayer.

**Figure 3 ijerph-19-16275-f003:**
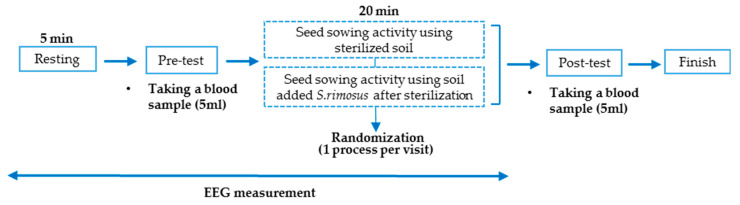
The experiment protocol. EEG: electroencephalogram.

**Figure 4 ijerph-19-16275-f004:**
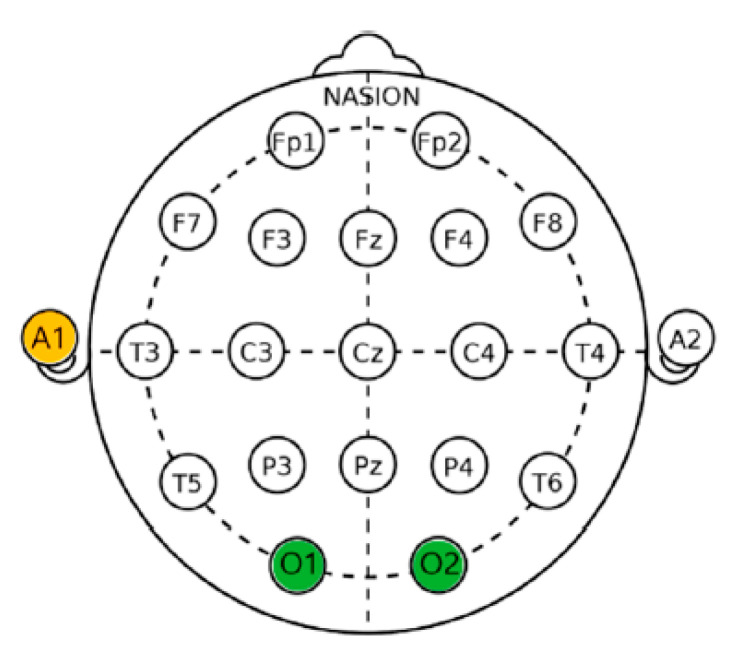
An international electrode arrangement [33].

**Figure 5 ijerph-19-16275-f005:**
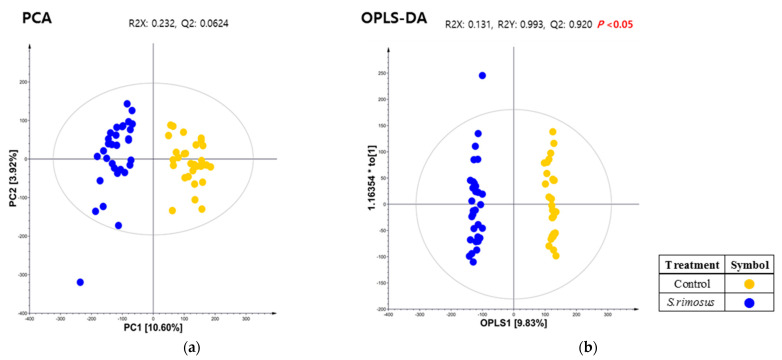
(**a**) The principal component analysis (PCA) and (**b**) orthogonal partial least square-discriminant analysis (OPLS-DA) score plot derived from GC-TOF-MS datasets for serum samples. Symbols, control (soil treated with media, ●) and treatment (*S. rimosus*, ●).

**Figure 6 ijerph-19-16275-f006:**
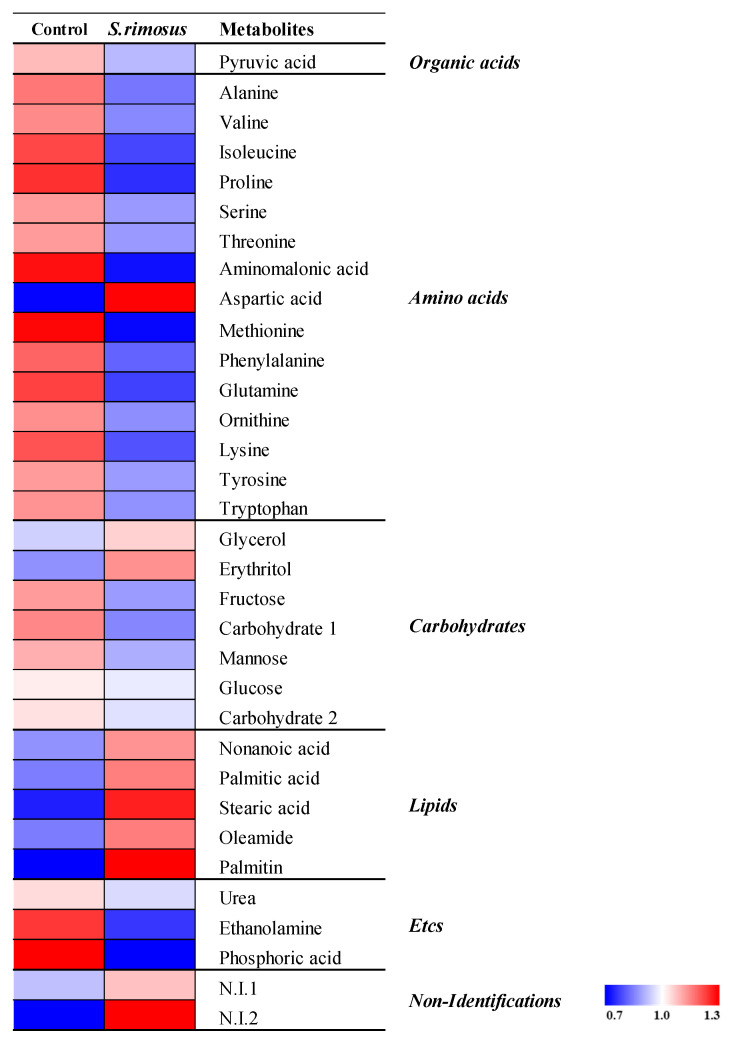
The heat map analysis for the relative abundance of different serum metabolites (VIP > 1.0) derived from GC-TOF-MS analysis. The colored squares (blue to red) indicate fold changes that were normalized by the average of each metabolite. Significantly different metabolites between control and treatment groups (*p* < 0.05, Student’s *t*-test).

**Table 1 ijerph-19-16275-t001:** Criteria and cautions for selecting/excluding research participants [30].

Selection criteria	A person who does not have psychopathological diseases and does not take related drugs A person who is right-hand dominant
Exclusion criteria	A person who does not agree to participate in the research even though he/she fully understands the contents of the research A person with a history of cardiovascular diseases such as high blood pressure, unstable angina, heart attack, and heart surgery A person with diseases such as olfactory function-related disorders, allergies, respiratory diseases, and insomnia A person who is pregnant, lactating, or menstruating
Requirements of participation	Stop drinking fluids the day before the experiment Prohibit excessive physical activity (e.g., breathless high-intensity physical activity for more than 60 min) on the day before the experiment Prohibit consumption of caffeinated beverages and smoking within 2 h before the experiment Not use cosmetics with strong scents such as perfumes and sprays on the day of the experiment

**Table 2 ijerph-19-16275-t002:** EEG power spectrum indicators used in this study.

Analysis Indicators	The Full Name of the EEG Power Spectrum Indicator	Wavelength Range (Hz)
RG	Relative power of gamma	(30–50)/(4–50)
RHB	Relative power of high beta	(20–30)/(4–50)
RST	Ratio of SMR to theta	(12–15)/(4–8)
RMT	Ratio of mid beta to theta	(15–20)/(4–8)
RSMT	Ratio of SMR—mid beta to theta	(12–20)/(4–8)
SEF50	Spectral edge frequency 50%	4–50
SEF90	Spectral edge frequency 90%	4–50

**Table 3 ijerph-19-16275-t003:** The clinical characteristics of the participants (N = 31).

Variable	Male	Female	Total
	Mean ± SD ^1^		
% (N)	29.0 (9)	71.0 (22)	100 (31)
Age (years)	27.13 ± 1.73	35.18 ± 11.39	32.66 ± 10.16
Height (cm)	172.93 ± 4.30	162.59 ± 5.42	165.94 ± 7.23
Body weight (kg)	75.31 ± 9.38	59.24 ± 8.39	64.27 ± 11.54
Body mass index (kg·m^−2^) ^2^	25.12 ± 2.52	22.45 ± 3.24	23.25 ± 3.23
SSS ^3^	76.0 ± 10.1	85.2 ± 10.1	82.2 ± 10.7
VAS ^4^	7.5 ± 0.9	7.7 ± 1.2	7.6 ± 1.1

^1^ SD, standard deviation; ^2^ Body mass index = Weight/Height^2^; ^3^ SSS = Scent survey for screening; A score of 74 or higher is a normal olfactory group [31]; ^4^ VAS = Visual analogue scale; A score of 5 or higher is a normal olfactory group [31].

**Table 4 ijerph-19-16275-t004:** The results of Electroencephalograph (EEG) according to the presence and absence of *Streptomyces rimosus (S. rimosus)* in the soil during the seed-sowing activity.

EEG (O1)		RG ^1^	RHB ^2^	RST ^3^	RMT ^4^	RSMT ^5^	SEF50 ^6^	SEF90 ^7^
		Mean ± SD ^8^						
Total (N = 31)	Using sterilized soil (control)	0.29 ± 0.07	0.18 ± 0.03	0.41 ± 0.12	0.57 ± 0.20	0.98 ± 0.32	18.61 ± 4.50	41.77 ± 3.21
	Using soil added *S. rimosus* after sterilization	0.32 ± 0.05	0.19 ± 0.02	0.48 ± 0.12	0.68 ± 0.20	1.16 ± 0.31	20.53 ± 3.12	43.07 ± 1.19
	Significance ^9^	0.062 ^NS^	0.058 ^NS^	0.046 *	0.036 *	0.038 *	0.073 ^NS^	0.057 ^NS^
EEG (O2)		RG	RHB	RST	RMT	RSMT	SEF50	SEF90
		Mean ± SD						
Total (N = 31)	Using sterilized soil (control)	0.28 ± 0.06	0.18 ± 0.03	0.41 ± 0.10	0.56 ± 0.17	0.97 ± 0.27	18.12 ± 3.81	41.76 ± 2.02
	Using soil added *S. rimosus* after sterilization	0.31 ± 0.04	0.19 ± 0.02	0.49 ± 0.10	0.68 ± 0.17	1.17 ± 0.26	20.16 ± 2.82	42.84 ± 1.08
	Significance	0.024 *	0.047 *	0.003 **	0.003 **	0.003 **	0.026 *	0.018 *

^1^ RG power spectrum was calculated by [gamma (30 to 50 Hz) power]/[total frequency (4 to 50 Hz) power.]; ^2^ RHB power spectrum was calculated by [high beta (20 to 30 Hz) power]/[total frequency (4 to 50 Hz) power.]; ^3^ RST was calculated by [SMR (12 to 15 Hz) power]/[theta (4 to 8 Hz) power.]; ^4^ RMT was calculated by [mid beta (15 to 20 Hz) power]/[theta (4 to 8 Hz) power.]; ^5^ RSMT was calculated by [SMR ~ mid beta (12 to 20 Hz) power]/[theta (4 to 8 Hz) power.]; ^6^ SEF50 is the spectral edge frequency 50%; ^7^ SEF90 is the spectral edge frequency 90%; ^8^ SD, standard deviation; ^9^ Statistical significance as determined using a paired t-test; O1 = left occipital lobe; O2 = right occipital lobe; NS, *, ** Nonsignificant, significant at *p* < 0.05 or significant at *p* < 0.01, respectively.

## Data Availability

The datasets generated for this study are available on request to the corresponding author.

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
