# Peer review of "Psychophysiological Responses of Humans during Seed-Sowing Activity Using Soil Inoculated with Streptomyces rimosus"

_ijerph, 2022, doi:10.3390/ijerph192316275_

Round 1

Reviewer 1 Report

Dear Authors and Editors,

I read with great attention the article sent for review entitled Physiological and Metabolomic Responses in Adults during Seed-Sowing Activity Based on Streptomyces rimosus Inoculated soil. The problem addressed by the authors related to the issues of developing new methods of stress reduction is interesting and timely. The results of published studies indicate that volatile metabolites generated from microorganisms in soil are introduced through the respiratory tract and can affect changes in human psychophysiology and biological metabolites. As the authors write, in order to clearly understand the healing mechanism of horticultural and nature-based complementary and alternative therapies, it is necessary to study the effect of this comprehensive approach and the role of each plant and soil. The goal of this project was to study the psychophysiological and metabolomic responses of adults during a horticultural activity (sowing seeds) using soil inoculated with the gram-positive bacterium - Streptomyces rimosus.

As the article reads, in this study, electroencephalogram (EEG) responses and serum metabolite levels were used to examine the effects of a horticultural activity (seed sowing) on adult psychophysiological aspects based on the presence or absence of the soil microorganism Streptomyces rimosus. The study was conducted on a very small group of people, namely 31. Of course, from the point of view of the correctness of statistical research, this sample is sufficient, but in my opinion too small to confirm the hypotheses presented in the article. Another doubt relates to the choice of participants for the experiment. What was the mental state of the participants, were they under stress in the period preceding the experiment, did they have a history of past depressive episodes, what is their average cortisol level? Were there rich and poor people, or only poor people among the surveyed group of people? (Usually, poor people participate in these types of projects because they receive a fee). This article raises serious doubts in my mind about the validity of the inference. I admit that the results are surprising to me, usually in this type of research inference is difficult and the results are unclear, which is why such studies are repeated, for example, at different times of the year or on different ethnic groups of participants. Participants' behavior can be influenced by many different factors, including the fee received for participating in research. Control studies should be a bit more elaborate to exclude the possible influence of other factors. Was the research conducted on all participants at the same time of day? The human reaction usually varies at different times of the day, we react differently in the morning and in the evening, and also differently after a meal. In this article, the results are almost perfect - that's surprising.

And the most important point is the soil is never sterile, it contains a huge number of microorganisms and fungi, and we often observe synergistic effects of co-occurring microorganisms, such as on mammals seeking food. Inoculation of soil with Streptomyces rimosus proves nothing. Streptomyces are a group of soil bacteria that are the source of most antibiotics in clinical use today. Streptomyces rimosus is the best-known filamentous soil bacterium as a primary source of antibiotics from the tetracycline class, especially oxytetracycline, which has been widely used against many gram-positive and gram-negative pathogens and protozoan parasites. Inoculation of soil with this bacterium may affect interactions with other important microorganisms. Detailed microbial profiles of this soil would need to be done to see what other microorganisms might have similar effects. Did the authors take this into account?

In my opinion, this article is of low quality, contains many serious methodological errors, and should be rejected in its current version.

Author Response

Point 1:

As the article reads, in this study, electroencephalogram (EEG) responses and serum metabolite levels were used to examine the effects of a horticultural activity (seed sowing) on adult psychophysiological aspects based on the presence or absence of the soil microorganism Streptomyces rimosus. The study was conducted on a very small group of people, namely 31. Of course, from the point of view of the correctness of statistical research, this sample is sufficient, but in my opinion too small to confirm the hypotheses presented in the article. Another doubt relates to the choice of participants for the experiment. What was the mental state of the participants, were they under stress in the period preceding the experiment, did they have a history of past depressive episodes, what is their average cortisol level? Were there rich and poor people, or only poor people among the surveyed group of people? (Usually, poor people participate in these types of projects because they receive a fee). This article raises serious doubts in my mind about the validity of the inference. I admit that the results are surprising to me, usually in this type of research inference is difficult and the results are unclear, which is why such studies are repeated, for example, at different times of the year or on different ethnic groups of participants. Participants' behavior can be influenced by many different factors, including the fee received for participating in research. Control studies should be a bit more elaborate to exclude the possible influence of other factors. Was the research conducted on all participants at the same time of day? The human reaction usually varies at different times of the day, we react differently in the morning and in the evening, and also differently after a meal. In this article, the results are almost perfect - that's surprising. The most important point is the soil is never sterile, it contains a huge number of microorganisms and fungi, and we often observe synergistic effects of co-occurring microorganisms, such as on mammals seeking food. Inoculation of soil with Streptomyces rimosus proves nothing. Streptomyces are a group of soil bacteria that are the source of most antibiotics in clinical use today. Streptomyces rimosus is the best-known filamentous soil bacterium as a primary source of antibiotics from the tetracycline class, especially oxytetracycline, which has been widely used against many gram-positive and gram-negative pathogens and protozoan parasites. Inoculation of soil with this bacterium may affect interactions with other important microorganisms. Detailed microbial profiles of this soil would need to be done to see what other microorganisms might have similar effects. Did the authors take this into account?

Response 1:

Dear reviewers,

We appreciate the time and effort that reviewers have dedicated to providing valuable feedback on my manuscript. We are grateful to the reviewers for their insightful suggestions on our paper. We have revised the manuscript upon your suggestions, which has been marked in blue for your review.

First, the purpose of this study was not to evaluate the effect on the psychopathological purpose, but to measure the biometric data of the psychophysiological aspect and examine the trend. In addition, since there has been no program study using soil microorganisms for humans to date, this study was conducted on general adults without a history of psychopathology as a basic study for a new trial (Lines 98-100). The exclusion criteria used for recruiting subjects were added to the Line 112. Recruitment documents were attached to the library, classroom, etc. in the university through convenient sampling method to make the recruitment range constant, and the average age was 33 years old. As pointed out by the reviewer, although not all subjects participated in the experiment at the same time, they were instructed to fast for 2 hours prior to EEG measurement and blood collection, and the experiment environment was created by maintaining constant temperature and humidity in the experiment space (Lines 119-121). In addition, all the soils used in the experiment (control soil, experimental soil) were provided to the subjects in a random crossover experiment. However, in agreement with the reviewer's opinion, future research will proceed with recruitment using a more sophisticated method to exclude various factors depending on the subject.

Additionally, the soil used in the experiment was 100% sterile soil inoculated with the soil microorganism S. rimosus. As noted by the reviewer, this study did not profile new microorganisms upon mixing, but this is another area of budgetary research and was not considered essential for the purpose of this study. However, I strongly agree that it will be a better study if these data are added, and I will try a future study after the research budget is secured. As for the deficiencies in the research methods you mentioned, i acknowledge that this study is an initial trial and is not designed as a complete experiment, so there are limitations in interpreting the results. We plan to revise the design in future studies and validate the findings in a more specific way. This was rewritten and emphasized in the conclusion part (Lines 354-359).

Again, we genuinely appreciate your attention and suggestions on this article.

Reviewer 2 Report

This is a good study with some significant applications to the study of horticultural therapy horticulture in general. I only have a few suggestions for improving the manuscript. 

Line 51-52 - It may be necessary to point out that the study of [17] focused on mice and not humans. It is recognized that initial studies of animals can and do translate over to human interventions. 

Line 90 - I believe this is initial exploratory research so a convenience sample is ok here. But I think in your conclusions need to highlight the need for a random sampling methos for future study to ensure the results can be generalized to a larger population.

Line 94 - It would have been helpful to see a comparison of right and left dominant participants. This may be a good analysis for a future study. 

2.5 experimental protocol Lines 138-150 - I noticed in the photos that the participants did not have gloves on. Many people garden and wear gloves, please describe the reasoning for not wearing gloves.  Also, the mixing of the soil was done by hand. Using a hand trowel may have also had different results. Both gloves and trowels could add to a future study and provide better recommendations for horticultural therapy interventions. 

2.5 experimental protocol Lines 138-150 - You don't mention any hand washing procedures prior to or after the session? I would discuss that procedure as part of your overall protocol. I am wondering if this is maybe a limitation. If you had participant wash hands, did the soap have scent in it and pose a confounding variable. 

Lines 353-357 - see comments above for possible additional limitations of this study.

Author Response

Dear reviewers,

We appreciate the time and effort that reviewers have dedicated to providing valuable feedback on my manuscript. We are grateful to the reviewers for their insightful suggestions on our paper.

We have revised the manuscript upon your suggestions, which has been marked in red for your review.

Point 1:

Line 51-52 - It may be necessary to point out that the study of [17] focused on mice and not humans. It is recognized that initial studies of animals can and do translate over to human interventions.

Response 1:

In agreement with the reviewer's suggestions, a sentence was added to the introduction, pointing out the limitations of previous studies (Lines 52-55).

Point 2:

Line 90 - I believe this is initial exploratory research so a convenience sample is ok here. But I think in your conclusions need to highlight the need for a random sampling methos for future study to ensure the results can be generalized to a larger population.

Response 2:

We agree with the reviewer's comments. In the conclusions part, we added emphasis on population expansion and random sampling in future studies (Lines 360-362).

Point 3:

Line 94 - It would have been helpful to see a comparison of right and left dominant participants. This may be a good analysis for a future study.

Response 3:

We appreciate the reviewer's comment. In the conclusions part, we added for possible limitations of this study (Lines 362-364).

Point 4:

2.5 experimental protocol Lines 138-150 - I noticed in the photos that the participants did not have gloves on. Many people garden and wear gloves, please describe the reasoning for not wearing gloves.  Also, the mixing of the soil was done by hand. Using a hand trowel may have also had different results. Both gloves and trowels could add to a future study and provide better recommendations for horticultural therapy interventions.

Response 4:

Thank you for your valuable comment. This study is a basic exploratory study conducted as a program experiment using soil inoculated with soil microorganisms. According to previous studies that investigated the inflow path of soil microorganisms, the experiment was designed based on the fact that it was mainly made through respiratory and skin contact (Lines 47-49). However, in actual horticultural sites, there are many cases in which activities are performed wearing gloves and a trowel rather than bare hands, so it is agreed that additional experiments need to be conducted.

Point 5:

2.5 experimental protocol Lines 138-150 - You don't mention any hand washing procedures prior to or after the session? I would discuss that procedure as part of your overall protocol. I am wondering if this is maybe a limitation. If you had participant wash hands, did the soap have scent in it and pose a confounding variable.

Response 5:

Thank you for your valuable comment. In this study, hands were washed before and after the experiment for the purpose of preventing the COVID-19 virus and blocking other germs. However, the soap used for hand washing was commercially available unscented soap, so the variable for fragrance was controlled.

Point 6:

Lines 353-357 - see comments above for possible additional limitations of this study.

Response 6:

As answered above, the limitations of this study and suggestions for future research have been added (Lines 360-364).

Again, we genuinely appreciate your attention and suggestions on this article.

Round 2

Reviewer 1 Report

The authors' changes in the manuscript's text are minor. Although the authors expanded the methodological part of the manuscript, added details about the selection of participants in the experiment, I still think that this is the weakest part of the manuscript. The results presented by the authors are not reliable in my opinion. Also, the interpretation of these results shows little knowledge of the authors of the paper, especially about the biology of soil microorganisms. In my opinion, this article after corrections is still not suitable for publication.

Author Response

Point 1:

The authors' changes in the manuscript's text are minor. Although the authors expanded the methodological part of the manuscript, added details about the selection of participants in the experiment, I still think that this is the weakest part of the manuscript. The results presented by the authors are not reliable in my opinion. Also, the interpretation of these results shows little knowledge of the authors of the paper, especially about the biology of soil microorganisms. In my opinion, this article after corrections is still not suitable for publication.

Response 1:

Dear reviewer,

We are grateful to the reviewer for your comments on our paper.

Soil microorganisms have been known as a major component of anticancer drugs used in the human body, along with various studies on their effects on plant growth. However, research on the contact with soil microorganisms or the reaction of fragrance components produced therefrom to the human body is very lacking. It is also fully recognized that interactions with microorganisms change the soil environment. Again, this study is a very early attempt to investigate the effects of fragrance components on the human body by a single soil microorganism, and it is a study that sees its potential. At this stage, Streptomyces rimosus used in this study was identified as a microorganism that releases a large amount of geosmin and 2-methylisoborneol (2-MIB), and the research method was designed based on the results of previously published studies (Kim et al., 20221). Also, a study measuring the effect of olfactory stimulation of these components on the psychophysiological responses of adults showed that alpha wave activity increased in the brain's frontal lobe, which positively affects emotional stability (Kim et al., 20172). However, the necessity of this study is emphasized by distinguishing it from the previous study by performing seed-seeding activity consisting of four movements during 20 minutes, beyond the short-term intervention time of 5 minutes mentioned as a limitation in the previous study. In the future studies, we plan to measure the effects of more diverse types of soil microorganisms or complex soil microorganisms, and plan to further supplement and verify the subject selection and research methodological aspects. In addition, for the past 15 years, this research team has been leading internationally in research on the effects of horticultural activities and horticultural treatment programs on human health. As we have published the largest number of research papers in this field worldwide, we are now approaching soil microbe factors among the factors that give health effects of horticultural therapy. These attempts are considered very meaningful in our field, and the interest of related researchers is also very high. With reference to your comments, we will continue our research in this field in more depth. Again, we thank you for the review.

1 Kim, S. O., Kim, M. J., Choi, N. Y., Kim, J. H., Oh, M. S., Lee, C. H., & Park, S. A. (2022). Psychophysiological and Metabolomics Responses of Adults during Horticultural Activities Using Soil Inoculated with Streptomyces rimosus: A Pilot Study. International Journal of Environmental Research and Public Health, 19(19), 12901.

2 Kim, M.; Sowndhararajan, K.; Kim, T.; Kim, J.E.; Yang, J.E.; Kim, S. Gender differences in electroencephalographic activity in response to the earthy odorants geosmin and 2-methylisoborneol. Appl. Sci. 2017, 7(9), 876.